# An Application of Uncertainty Quantification to Efficiency Measurements and Validating Requirements through Correlating Simulation and Physical Testing Results [note 1]

**DOI:** 10.3390/s24154867

**Published:** 2024-07-26

**Authors:** Michael Leighton, Uday Akasapu

**Affiliations:** 1AVL List GmbH, 8020 Graz, Austria; 2BET-Motors GmbH, 8020 Graz, Austria; uday.akasapu@bet-motors.com

**Keywords:** validation, uncertainty quantification, efficiency measurement, simulation, Bayesian

## Abstract

Validation is a critical aspect of product development for meeting design goals and mitigating risk in the face of considerable cost and time commitments. In this research article, uncertainty quantification (UQ) for efficiency testing of an Electric Drive Unit (EDU) is demonstrated, considering confidence in simulations with respect to the validation campaign. The methodology used for UQ is consistent with the framework mentioned in the guide to the expression of uncertainty in measurement (GUM). An analytical evaluation of the measurement chain involved in EDU efficiency testing was performed and elemental uncertainties were derived, later to be propagated to the derived quantity of efficiency. When uncertainties were associated with measurements, the erroneous measurements made through sensors in the measurement chain were highlighted. These results were used for the assessment of requirement coverage and the validation of test results.

## 1. Introduction

A typical EDU used in a Battery Electric Vehicle (BEV) is a compact unit comprising the electric motor, power electronics, and transmission, often fully integrated into one unit. As shown in Figure 1, a highly integrated EDU requires a few necessary, conflicting elements that must be balanced within the requirements. Based on the minimum targets defined in the vehicle architecture, the EDU must have features such as high efficiency and high specific power, small package dimensions and low total weight, low current consumption, economical mass production, scalability, adequate Noise Vibration and Harshness (NVH) behaviour, and integrated cooling [1]. Due to the desire for greater driving range in BEVs and the resulting impact on battery size and cost, system efficiency is a crucial aspect of EDU development.

The validation of a requirement is dependent on the collection of sufficient evidence to determine if the requirement is fulfilled or not. Within this process, evidence collected from physical testing is typically and traditionally the contribution being considered. Often, the consideration of measurement uncertainty is overlooked in this process, disregarding the confidence in the resultant validation of the corresponding requirement. In an alternative to the quantification of measurement uncertainty, validators sometimes rely on repeating measurements with a selection of samples in which all must pass the test criteria or the requirement threshold. Approaches with a subset of samples have the benefit of increasing the confidence associated with the result in correspondence with the binomial law, often applied in reliability calculation, but, without consideration of the result uncertainty, it is not quantified and a predefined number of test samples could prove insufficient or wastefully superfluous. Another alternative is to consider the validation of a full application, such as a BEV, as a series of sub-system validations, such as the transmission, e-motor, battery, inverter, chassis, etc. In such an approach, the final homologation of the application becomes a final validation of all contributing sub-systems in a V-Model approach (Figure 2) [3]. While this approach can be considered to save the overall validation efforts, it clearly introduces the risk that the final product will fail to meet the requirements and that the evidence for this could remain undiscovered until the late stages of the development when a significant investment has been wasted. The efforts to correct any late-stage issue will prove significant and, therefore, the risk is often considered unacceptable. This aspect can lead to additional integration testing and tests intended to validate the system model applied in the requirement decomposition to the left of the V-Model.

Another aspect that is commonly neglected in the validation of requirements is the impact of simulation results. Simulations are regularly applied in the development cycle, providing invaluable feedback and development guidance, without which advanced design would not be possible. The associated simulations in the development process are trusted to provide guidance and feedback to the development, on the basis of which designs are modified or frozen. The simulations taking place are typically guided by the requirements that drive the design process and must later be validated both by Failure Mode Effect Analysis (FMEA) [4] and expert experience. However, these same simulations are not commonly included in the validation process directly. One major hurdle in the inclusion of simulation-based validations is the definition and application of the associated uncertainty in a simulation result. In addition to the measurement uncertainty, typically overlooked for convenience, the results and uncertainty in simulation results can be included and considered for their contribution to the validation process.

This paper presents a study into the validation of EDU efficiency as an extension of an earlier publication [5], both in terms of physical testing with the associated uncertainty in the measurement chain and the contribution and combination of simulation results, considering the associated uncertainty in simulation results. The methodology was validated with an empirical approach, taking a priori uncertainty distributions and a large sample of randomly populated potential results to demonstrate the associated confidence in the result. The inclusion of simulation results and their contribution to confidence will be shown to be a major factor in the validation of a design.

While other authors have considered the uncertainty contributions in testbed measurement systems [6], the implication on the validation confidence has not previously been considered for significant requirements, such as efficiency targets. Additionally, the contribution to result confidence resulting from simulations or combined sources of evidence has previously been discussed [7,8], however, without quantifying the uncertainty contributions from the sensors employed.

## 2. Materials and Methods

### 2.1. Technical Background

Efficiency is an essential parameter to understand the ability of the EDU to convert from primary High-Voltage Direct Current (HVDC) energy (electrical energy) to productive propulsion (mechanical energy). The HVDC is often in the region of 400 to 800 V as a nominal battery supply and is distinguished from the the High-Voltage Alternating Current (HVAC) supplied to typical electric motor designs and the low voltage used in the power electronics for controls and auxiliaries. Improvements in the efficiency of all main and auxiliary systems have a direct impact on energy and operational expenses, lowering the Total Cost of Ownership (TCO). The TCO for battery electric passenger cars was shown to fall below that of Internal Combustion (IC) vehicles after a period of approximately 7 years in 2021 [9], and is constantly improving with technological developments. As 7 years is well below the expected lifetime of many vehicles [10], there is a clear benefit for the end users. Similarly, studies have shown that battery electric Commercial Vehicles (CVs) are expected to transition to having a lower TCO than their IC counterparts from around 2025 [11] with the possibility of supportive funding to accelerate and encourage the transition [12,13]. The value for the end customer lies in having more driving time on the road and less time at the charging station, particularly for CV applications. The TCO assessment on passenger car applications [9] excluded the resale and depreciation effects; however, commercial vehicle studies have concluded that battery efficiency degradation has a minimal impact on the TCO [13].

#### 2.1.1. Standard Cycles

To measure the electric energy consumption and electric range of a BEV, the vehicle was run on a dynamometer using driving cycles like the Worldwide harmonised Light vehicles Test Procedure (WLTP) [14]. WLTP is based on a standard testing method that was adopted by the United Nations Economic Commission for Europe (UNECE) in early 2014 [15]. Similar cycles are used in other regions, such as the Federal Test Procedure, FTP-75 [16] in the United States of America and the China Light-duty vehicle Test Cycle (CLTC) in China [17]. The size of the battery used in a BEV is therefore related to the vehicle range as a result of the performance of an EDU over the driving cycle. Efficiency, as a consequence, plays an important role in driving range testing and also in achieving low cost per kilometre of driving [18].

#### 2.1.2. State of the Art

For the evaluation of the efficiency of the EDU, suitable measuring and testing systems were required. In addition, knowledge of uncertainties was essential for evaluating the quality of the results obtained from the testbed. The Degree of Goodness (DoG) of a measurement of an experimental result can be described by the concept of uncertainty. For testing and calibration laboratories, ISO/IEC 17025 [19], General requirements for the competence of testing and calibration laboratories, has become significant in the demonstration of competence in the market and the ability to demonstrate capabilities of providing a customer with trustworthy results from measurements [20]. As a consequence, there is an increased emphasis on measurements and uncertainty quantification in the field of automotive testing.

As per ISO/IEC 17025, a procedure to estimate uncertainties of a measurement is necessary for any calibration laboratory or a testing laboratory that performs its own calibrations [19]. In essence, the ISO/IEC 17025 standard is applicable to any testing facility and chapter 7.6 demonstrates the process for such testing. All significant contributions, including those coming from sampling, must be taken into consideration when evaluating measurement uncertainty using proper analysis methods. For the expression of uncertainty in measurement, the ISO standard makes reference to the ISO/IEC Guide 98-3 [21].

JCGM 100 [22], Evaluation of measurement data—Guide to the expression of uncertainty in measurement, is the basis for all the guides for uncertainty measurement. The Joint Committee for Guides in Metrology (JCGM) created this document to promote the expression and evaluation of measurement uncertainty. ISO reissued the guide to the expression of uncertainty in measurement (GUM) as a standard with the name ISO/IEC Guide 98-3, Uncertainty of measurement—Part 3: Guide to the expression of uncertainty in measurement [21].

The American Society of Mechanical Engineers (ASME) has defined a standard for the evaluation of uncertainties in measurements, Test Uncertainty PTC 19.1 [23]. PTC, which stands for Performance Test Codes, has been produced to ensure accuracy, precision, and reliability in tests conducted by both manufacturers and the users of the equipment [24]. PTC 19.1 is a general document on analytical techniques for measurements which follows ISO/IEC Guide 98-3. ASME has 48 PTCs, which cover applications in the field of fluid handling, power production, combustion, heat transfer, and emissions. ASME further established a new set of standards that makes use of uncertainty quantification in order to verify and validate processes related to computational models and simulations. The ASME verification, validation, and uncertainty quantification (VVUQ) provide guidelines to verify the computational models created for the following purposes [25];

Verification: it is determined whether the computational model matches the mathematical description;Validation: it is applied to find out if the model accurately depicts the application in the real world;Uncertainty quantification: it is carried out to understand how changes in the numerical and physical parameters affect the simulation result.

ISO 17025 states requirements that allow laboratories that perform calibrations or tests to stay competent, impartial, and consistent with their activities of experimenting in the laboratories. A main focus of the standard is on the implementation of a management system, synchronous with the requirements in ISO 9001 [26]. Figure 3 shows an overview of the structure/content of ISO 17025.

#### 2.1.3. Efficiency Measurement in EDU

The test and measurement instruments, as shown in Figure 4, are products for electric power measurement, signal conditioning, data acquisition, and testbed management systems. The AVL testbed management system, PUMA (Prüfstands und Messtechnik Automatisierung), applies automation and controlling, and finishes the job of test execution on the testbed. A torque transducer by HBM was used for torque measurements. The torque transducer can also measure speeds but a separate speed transducer can also be used, as shown in the block diagram. Only DC voltages and currents were necessary for overall EDU efficiency as DC voltage and currents were measured for the input power determination. Torque and speed, measured in frequency signals, were fed to PUMA through a data acquisition system (DAQ) [27], which acted as a front-end module. A current transducer measured the current output using its primary circuit and the converted current in the secondary circuit was picked up by the Current Sensor Supply (CSS) box. Current measured through the current transducer was thus fed to the CSS box. The voltage was measured by a High-Voltage Probe (HVP). Both the CSS and HVP did the job of signal conditioning and then X-ion, the power analyser, completed the job of high-speed data acquisition from these signal conditioning units.

#### 2.1.4. Literature Review

Following the right procedures for the estimation of uncertainty and ensuring trustworthiness in measurement results are the key challenges in uncertainty quantification. Different methods for ensuring trustworthiness, comparison of data within the laboratory and inter-laboratory, and methods to monitor the results and improve their quality are discussed in the journal article by Charki and Pavese [28]. Castrup, [29] compares different tools or software applications that are used for analysing measurement uncertainty. Tools are compared on the basis of methods used for estimation, ease of use, functions that they offer, and the technical support available.

A quick reference guide by Hogan [30] simplifies the process of uncertainty quantification in eight steps, as seen in Figure 5. The first step instructs to choose the measurement function which has to be evaluated and define the measurement equipment to be used, the measurement procedure, and the measuring range of the measurand. Step two is to determine the various sources of measurement error. Users should evaluate the standards applicable to the measurement process and investigate existing information like manuals, publications by metrology institutes, or technical guides to find these sources of uncertainty. Step three describes information on how to quantify the components of uncertainty, in regards to the measurand. Step four expresses the concept of characterizing the identified sources of uncertainty into Type A and Type B. Type A uncertainty is obtained through the application of statistics to repeated measurements. When the estimation of uncertainty is made without any use of repeated measurements, it is called a Type B uncertainty. To find Type A uncertainty, the probability distribution of measured data is used and the distribution function is identified through the histogram created by the data set. Step five explains the method to convert uncertainty to standard deviations and, consequently, each source of uncertainty will be at the one σ level, where σ is the standard deviation. In step six, methods to combine the different uncertainties are explained and, in step seven, the estimated uncertainty is multiplied with a coverage factor to find the expanded uncertainty. Step eight is an extra step, executed after the calculation of expanded uncertainty to evaluate the contributions of different sources of uncertainty. A comparison can be made with data from different labs and the evaluated contributions are used to analyse the influence of a single source on the total measurement uncertainty.

The details covered by Ratcliffe and Ratcliffe [31] include various methods presented in the standards for uncertainty in measurement and explanation of the terminologies and the topics related to them. The authors expand on the topic of uncertainty budget, which is intended to be an easy-to-read table that shows the analysis of the different standard uncertainties that propagate and combine to become the standard uncertainty for a single measured quantity. This uncertainty budget allows a user to smartly identify the highest contributing uncertainty source and where to focus to make improvements. The authors also provide simple instructions for finding the standard uncertainty for a single measured quantity by combining several standard uncertainties using the below-mentioned formulae. First, finding combined standard uncertainties for Type A and Type B can be done as follows:(1)$$\begin{array}{ccc} \; & \; & \mathrm{Combined}\mathrm{Standard}\mathrm{Uncertainty}\mathrm{for}\mathrm{Type}A,\;u_{A}={\left\{\sum_{i}u_{A,i}^{2}\right\}}^{1/2} \end{array}$$
(2)$$\begin{array}{ccc} \; & \; & \mathrm{Combined}\mathrm{Standard}\mathrm{Uncertainty}\mathrm{for}\mathrm{Type}B,\;u_{B}={\left\{\sum_{i}u_{B,i}^{2}\right\}}^{1/2} \end{array}$$

These combinations are done using the GUM-recommended method of Root Sum of the Squares (RSS) of the individual standard uncertainties. The combined standard uncertainty of any measurement with Type A and Type B uncertainties is calculated as follows:(3)$$\begin{array}{ccc} \; & \; & \mathrm{Standard}\mathrm{Uncertainty}\mathrm{of}\mathrm{the}\mathrm{Measurement},\;u_{m}={\left\{u_{A}^{2}+u_{B}^{2}\right\}}^{1/2} \end{array}$$

For any measurement, the final value of uncertainty is always the expanded uncertainty, which is calculated as follows:(4)$$\begin{array}{ccc} \; & \; & \mathrm{Expanded}\mathrm{Uncertainty}\mathrm{of}\mathrm{the}\mathrm{Measurement},\;U_{m}=k.u_{m} \end{array}$$
where $u_{A,i}$ and $u_{B,i}$ are the individual uncertainties that fall under the category of Type A and Type B, respectively and *k* is the multiplying factor called the coverage factor for different confidence levels. Table 1 gives the values of the coverage factors. The coverage factor, *k*, can be determined from Student’s T-distribution, using the probability and the number of degrees of freedom. The probability in this case is 1 minus the confidence target. The degrees of freedom are typically set to infinity when they are not known. The 95% confidence level is commonly used unless otherwise stated (coverage factor of 1.96). ISO/IEC 17025 applies a coverage factor of 2 (95.45% confidence) [19].

The standards mentioned in Section 2.1.2 have information on the techniques and methods for uncertainty quantification while some researchers offer alternative approaches and demonstrate examples with respect to their field of work. Such studies with relevance to efficiency measurements or measurands that contribute to the derivation of efficiency were also considered.

Yogal et al. [32] focused on finding the difference between calculating measurement uncertainty for direct and indirect efficiency measurements in terms of time consumed to perform the measurements and the values of uncertainty. It was concluded that the indirect efficiency measurement method has a lower measurement uncertainty than the direct efficiency measurement method for determining the efficiency of the 7.5 kW Permanent Magnet Synchronous Machine (PMSM). In the direct efficiency measurement method, electrical input power and mechanical output power are measured, various dependent and independent variables affecting the measurement are assessed, and then GUM is used to calculate the associated measurement uncertainty for each of the variables. In the indirect efficiency measurement, modelled equations for the motor are used to calculate the electrical power and the different losses present in the motor and then the measurement uncertainty is evaluated. The direct efficiency evaluation method has more risks of high associated uncertainties at lower speed and torque values since the measurement instruments have a higher percentages of uncertainty at the lower limits of the measurement range.

Bucci et al. [33] discuss methods to determine uncertainty for induction motors below 5 kW. Direct and indirect efficiency calculations are followed for the examples showcased and the authors use GUM as well as IEC 60034-2-1 [34] for these calculations. Since the direct method accuracy is significantly influenced by the torque and speed sensor accuracies, it can benefit from the improving performance of torque transducers currently on the market and, thus, the authors propose this technique for induction motor uncertainty determination.

Beccherelli et al. [35] expand on the procedures mentioned in the standard, EN 50598-2, for determining uncertainty and provide guidelines on how the standard can be more useful for testing laboratories. EN 50598-2 is an obsolete standard and has been replaced by EN 61800-9-2 [36].

De Santis et al. [37] made a comparison of measured uncertainties from different journal articles for asynchronous motors and their own back-to-back method for a synchronous motor. Their research is a novel study on measurement uncertainties, as most studies previously had only made assessments on uncertainties for asynchronous electric motors.

Li et al. [38] considered efficiency tolerance, a value which has to be made available by manufacturers as per the standard, GB/T 755 [39]. The uncertainty is calculated using JJF 1001 [40] and JJF 1094 [41], which has a similar approach to GUM. As per their study, the values of calculated measurement uncertainties are compared with the efficiency tolerances. If the value of the measured uncertainty is <33% of the value of efficiency tolerance, the uncertainty is ignored in the efficiency evaluation process.

Song et al. [42] focus on the accurate measurement of the rotational speed in the efficiency evaluation process. Their method of determining rotational speed through the angle covered by the motor for a longer period of time yields a lesser uncertainty associated with mechanical power measurement and, thereby, with the total efficiency of the electrical machine.

Herman and Bojkovski [43] presented a comparison between the efficiency results obtained through measurements and the results obtained through a mathematical model; a direct-drive motor design is used here in the evaluation. The study does not show a direct comparison but rather focuses on the high-torque system for the appropriate testing of a direct-drive motor. This kind of system is found to be useful for the validation of the performance characteristics for a small batch of prototype motors.

Uncertainty quantification for efficiency measurement by the direct method of efficiency determination for Synchronous Reluctance Motors (SRMs) has been proposed by Kim et al. [44]. The work by Caruso et al. [45] investigates evaluating losses using differential measurements in a Power Drive System (PDS). As per IEC 61800-9-2 [36], a PDS is a system comprising of the electrical drive motor and the power electronics used to operate the motor, sometimes referred to as an e-drive. The authors in this study performed an uncertainty assessment of the power losses in order to accurately compare the power loss variations with different control strategies used to run the motor. The methods used for quantifying the uncertainty associated were as per IEC 60034-2-1 [34], which is applicable to all rotational electrical machines, excluding machines used for traction vehicles.

Dhakal et al. [46] made an analysis of uncertainty quantification for the efficiency measurements of a turbine used in hydroelectric power generation, but the procedure as well as a detailed breakdown are discussed in the bachelor thesis by Adhikari and Dhakal [47]. The entire calculation follows the GUM methodology and acts as an attractive source of information for dealing with measurement uncertainties.

The majority of the literature reviewed focused on non-traction applications, but made use of GUM or other relevant standards for uncertainty quantification, providing an overview of uncertainty quantification in efficiency measurement. For traction vehicles, Dong et al. [48] propose a method to quantify uncertainties for the efficiency testing of a drive motor system and analysed the sources that contribute to uncertainty.

Wegener and Andrae [49] put forward the sources of uncertainties for torque sensors as recommended by HBM and Minda [50] elaborates on the same topic. A study on the uncertainties for testbeds and possible sources of individual uncertainties has been conducted by Zivkovic et al. and Short [51,52]. Uncertainties were quantified as the brake-specific fuel consumption of an engine and measurement quantities of speed, torque, fuel consumption are involved in its determination.

Kowalak [53] demonstrated approaches for the uncertainty quantification of encoders and signifies the importance of errors associated with the measured value and consideration of the entire measurement chain for uncertainty quantification. A deep dive into uncertainty quantification for efficiency measurement of converted-fed induction motors was detailed by Kärkkäinen et al. [54].

### 2.2. Methods

#### 2.2.1. Associated Uncertainties with Efficiency Measurement

For every value of measurement, whether a primary quantity or derived quantity, the associated sources of uncertainty must be established and then uncertainty must be quantified. The uncertainty quantified depends on the accuracies certified by the equipment manufacturer or the calibration of the equipment. As per ISO 17025 [19], the calibration certificate of the measurement equipment should contain all the attained accuracies. A calibration lab can only try to achieve the mentioned accuracies of the equipment manufacturer. Hence, the uncertainties quantified based on the accuracy values mentioned in the specifications, for different effects like hysteresis, temperature effects or repeatability, become the minimum possible associated uncertainty of a measured value.

Within the EDU measurement system for efficiency validation, the following sources are evident:Torque measurement (torque transducer)Sensitivity tolerance, non-linearity and hysteresis, temperature effects on zero signal and span, repeatability, and parasitic loads.Speed measurement (rotational angular encoder)Signal sensitivity to temperature, non-linearity, and hysteresis.Current measurement (CSS)CSS accuracy and CSS amplifier error limitVoltage measurement (HVP)HVP accuracy and HVP amplifier error limitData Acquisition (AVL Xion)Absolute frequency error on sampling

The measurement uncertainty of a torque sensor can be defined in terms of the different elemental uncertainties that are associated with sensitivity, linearity and hysteresis, temperature effect on zero signal, temperature effect on full scale, repeatability, and the parasitic loads associated with the measurement. These contributions can be combined as an RSS to provide the torque measurement uncertainty. The uncertainty contribution from the speed sensors is calculated similarly.
(5)$$\begin{array}{ccc} \; & \; & \mathrm{Uncertainty}\mathrm{of}\mathrm{the}\mathrm{torque}\mathrm{measurement},\;U_{torque}=\sqrt{\sum u_{i}^{2}} \end{array}$$
(6)$$\begin{array}{ccc} \; & \; & \mathrm{Uncertainty}\mathrm{of}\mathrm{the}\mathrm{rotational}\mathrm{speed}\mathrm{measurement},\;U_{speed}=\sqrt{\sum u_{i}^{2}} \end{array}$$
where $u_{i}$ represents the individual contributing factors to the measurement uncertainty, as collected in Table 2 and Table 3.

Flux-compensated transducers require a certain amount of supply current, via which the magnetic field produced by the current-carrying measuring line is compensated. This produces a particularly linear behaviour. The transducer supply current is provided by the CSS-Box. The output signal of the flux-compensated transducers is generally, once again, a current signal (typically ≤ 1 A), which the CSS-Box converts into a voltage signal of 10 V amplitude, and, in turn, can be connected directly to a DAQ module. The accuracies of the CSS-Box must therefore be considered in the signal path in addition to the influence of the accuracy of the current transducers. The accuracy specification for the AVL CSS-Box alone is
(7)$$\begin{array}{ccc} \; & \; & {\mathrm{Accuracy}}_{\mathrm{CSS}-\mathrm{Box}}=\pm \left(0.005\%\mathrm{Reading}+0.02\%\mathrm{Range}\left(1A,0.4A,0.2A\right)\right)\; \end{array}$$

For a combination of the AVL DAQ connection module and CSS-Box, the following specification applies: (8)$$\begin{array}{ccc} \; & \; & {\mathrm{Accuracy}}_{\mathrm{DAQ}\mathrm{and}\mathrm{CSS}-\mathrm{Box}}=\pm \left(0.007\%\mathrm{Reading}+0.028\%\mathrm{Range}\left(1A,0.4A,0.2A\right)\right)\; \end{array}$$

High voltages are conditioned using the HVP-Box (integrated in the DAQ), which divides the applied HV signals in a ratio of 150:1 or 75:1, so that voltage ranges of 1500 V or 750 V each result in a maximum output voltage of 10 V. This is subsequently digitised by the DAQ. The AVL HVP-Box bandwidth is approximately 20 MHz. This enables a faithful reproduction of pulse-width modulated voltage signals in AC applications. The accuracy specification for the AVL HVP-Box is
(9)$$\begin{array}{ccc} \; & \; & {\mathrm{Accuracy}}_{\mathrm{HVP}-\mathrm{Box}(\mathrm{AC})}=\pm \left(0.015\%\mathrm{Reading}+0.02\%\mathrm{Range}\left(750V/1500V\right)\right)\; \end{array}$$

For DC measurements a slightly different input voltage range is applied: ±1200 V and ±600 V. The accuracy specification for the AVL HVP-Box is: (10)$$\begin{array}{ccc} \; & \; & {\mathrm{Accuracy}}_{\mathrm{HVP}-\mathrm{Box}(\mathrm{DC})}=\pm \left(0.02\%\mathrm{of}\mathrm{reading}+0.03\%\mathrm{of}\mathrm{range}\left(1200V/600V\right)\right)\; \end{array}$$

The AVL X-ion is used for the data acquisition with the X-FEM E4H2 connection for fast voltage signals. The X-FEM I4H2 has voltage ranges of 10 V and 1 V as well as current ranges of 1 A and 0.5 A and is used for acquisition of current signals from typical flux-compensated current transducers. The signals are captured with 18-bit resolution at a sampling rate of 2 MHz per channel. The module E4H2 provides the input voltage ranges ±60 V, ±10 V, and ±1 V. In conjunction with the HVP-Box and the CSS-Box, the ±10 V range is used. The accuracy specification for DC signals and sinusoidal AC signals in a frequency range up to 30 kHz for the 10 V range is: (11)$$\begin{array}{ccc} \; & \; & {\mathrm{Accuracy}}_{\mathrm{DAC}(I/V)}=\pm \left(0.005\%\mathrm{Reading}\left(V\right)+0.02\%\mathrm{Range}\left(10V\right)\right)\; \end{array}$$

At higher frequencies, the influence of the installed analog 800 kHz anti-aliasing filter becomes noticeable. Due to the 2 MHz sampling rate, the 800 kHz anti-aliasing filter is required (to avoid aliasing effects, the anti-aliasing filter frequency must be lower than half of the sampling frequency). The anti-aliasing filter is used to pass only the desired frequencies.

#### 2.2.2. Efficiency Measurement Uncertainty Analysis

The standard uncertainties of speed and torque are calculated first, where each standard uncertainty is calculated using the applicable contributing factors. The uncertainties of speed and torque do not correlate, as they are measured with separate sensors; hence, the uncertainty contributions of both quantities can be analysed individually and combined in RSS. Considering the test setup for an EDU with speed and torque on each side of the output, the uncertainty of each side is examined individually:(12)$$\begin{array}{ccc} \; & \; & \mathrm{Uncertainty}\mathrm{of}\mathrm{each}\mathrm{side}\mathrm{output}\mathrm{power},\;U_{power_{side}}=\sqrt{U_{speed}^{2}+U_{torque}^{2}} \end{array}$$

The requirements for any efficiency calculation test case are to identify the values necessary for determining the output power ($P_{output}$) and the input power ($P_{input}$). The output power of an electric drive system can be calculated as follows: (13)$$\begin{array}{ccc} \; & \; & \mathrm{Total}\mathrm{output}\mathrm{power},\;P_{output}=\frac{2\times \pi }{60}\times \left[\left(n_{output_{left}}\times M_{output_{left}}\right)+\left(n_{output_{right}}\times M_{output_{right}}\right)\right] \end{array}$$
where *n* is the speed in rpm and *M* is the torque in Nm from the output measurement, resulting in output power $P_{output}$ in W.

The combination of transducer elemental uncertainties gives the standard uncertainty of torque and speed measurements. Torque and speed are primary quantities and power is a derived quantity. Hence, the uncertainty of these primary quantities has to be propagated to the final quantity. These standard uncertainties of the power measurement must also be adjusted by the DAQ uncertainty in the signal capture. The total output power standard uncertainty therefore becomes
(14)$$\begin{array}{c} \;U_{Outputpower}=\sqrt{U_{power_{left}}^{2}+U_{power_{right}}^{2}+U_{DAQleft}^{2}+U_{DAQright}^{2}} \end{array}$$
where $U_{Outputpower}=$ Uncertainty of total output power

The standard uncertainties of voltages and currents were calculated as given in Section 2.2.1. In the same way as the uncertainty in power was calculated for the output side, the propagated uncertainty of input power was combined using the RSS method. The calculation of the uncertainty of voltage is a simple process. The measured voltages are picked up by the combination of HVP and Xion and a straightforward accuracy formula is used to determine the uncertainty. In the case of current measurement, due to the use of current sensors, first, the elemental uncertainties due to current sensors are calculated, and then, the accuracy formula for CSS and X-ion combination is applied to the final combined uncertainty of the current measurement. The current sensor final uncertainty is the RSS of full-scale error, secondary error, and CSS error:(15)$$\begin{array}{ccc} \; & \; & \mathrm{Uncertainty}\mathrm{of}\mathrm{total}\mathrm{input}\mathrm{power},\;U_{inputpower}=\sqrt{U_{HVvoltage}^{2}+U_{HVcurrent}^{2}} \end{array}$$

The efficiency of the EDU is calculated as the fraction of the input power that is converted into useful output power. In the case of the forward-driving operation of the EDU, this is simply given as:(16)$$\begin{array}{ccc} \; & \; & \mathrm{EDU}\mathrm{forward}-\mathrm{driving}\mathrm{efficiency},\;\eta =\frac{P_{out}}{P_{in}} \end{array}$$

Performing an RSS of the uncertainties on the input side and on the output side gives the final expanded uncertainty of the efficiency measured. The process of propagation of uncertainty is given, once again, by RSS:(17)$$\begin{array}{ccc} \; & \; & \mathrm{Uncertainty}\mathrm{in}\mathrm{efficiency},\;U_{efficiency}=\sqrt{U_{inputpower}^{2}+U_{outputpower}^{2}} \end{array}$$

#### 2.2.3. Combining Simulation Results with Physical Testing Results

The combination of simulation results and physical testing results can serve to increase the confidence associated with the result of the validation process. Additionally, as the simulations are typically conducted as a fundamental part of design development, the results can be evaluated for their contribution to requirement coverage, ideally to define the extent to which testing is required to contribute to the validation and, alternatively, then, to support the validation through inclusion of all results in the validation outcome.

Initially, an evaluation of simulation uncertainty distribution must be made. The result of a simulation can then be considered for the requirement coverage confidence. If confidence is not sufficient from the simulation alone, it can be decided if additional simulation will be executed or physical testing is needed. Additional simulation must be executed with an alternative method, e.g., for a flow rate, Computational Fluid Dynamics (CFD) might be applied in parallel to Smooth Particle Hydrodynamics (SPH) in parallel to an analytical calculation, as long as result uncertainties are known. However, it would not be appropriate to re-run an SPH with a random initial particle distribution and count the simulation result as additional, unless the specific effect of the particle distribution on the result is known. Similarly, running two CFD simulations with different software packages should not be counted as two separate simulation contributions. This is due to the propensity for systematic errors in similar computation methods. If this is fully known, then appropriate calibration of the model can be applied; however, it is assumed here that this is not known with sufficient certainty in advance.

If an additional source of validation is provided, a product distribution can be constructed to allow the evaluation of the resultant confidence. If the resultant combined validation confidence is insufficient, additional tests or simulations can be applied, and the product of the distributions extended until the confidence is achieved or the product is deemed to have failed the requirement threshold.

Let us assume that we have a requirement of 100 kW peak power for an EDU. We conduct a simulation of the system with a calibrated simulation tool that has a normal distribution of uncertainty and a standard deviation of result scatter of 1.8 kW. The simulation result gives a peak power of 101.6 kW. For the sake of this example, the distribution is assumed to be a highly simplistic and symmetric Gaussian probability density function (Figure 6):(18)$$\begin{array}{c} \phi =\frac{1}{\sigma \cdot \sqrt{2}}\cdot e^{-\frac{1}{2}\cdot {\left(\frac{x-\mu }{\sigma }\right)}^{2}} \end{array}$$

Taking the area under the curve in the region above the requirement value, the confidence of this simulation result validating the requirement can be determined.
(19)$$\begin{array}{c} \phi_{sim}\left(x\right)=\frac{1}{\sigma_{sim}\cdot \sqrt{2}}\cdot e^{-\frac{1}{2}\cdot {\left(\frac{x-\mu_{sim}}{\sigma_{sim}}\right)}^{2}} \end{array}$$

The mean ($\mu_{sim}$) can be applied as the result from the test. The probability density function can be converted to a cumulative frequency distribution up to a given value (in this case, the requirement value of 100 kW), by taking the integral of the probability density function above that point.
(20)$$\begin{array}{c} P_{sim}\left(req.passed\right)=\int_{req}^{\infty }\left(\phi_{sim}\left(x\right)\right)\cdot dx \end{array}$$

The resultant confidence of a pass can be determined as approximately 81%. Assuming a target of 90% confidence, this result alone is not sufficient to complete the validation.

A physical test could then be conducted and results in a peak power determined at 101.1 kW. The evaluation of the measurement chain uncertainty at the given operating conditions indicates a normal distribution of uncertainty and a standard deviation of result scatter of 1.0 kW (Figure 7).
(21)$$\begin{array}{c} \phi_{test}\left(x\right)=\frac{1}{\sigma_{test}\cdot \sqrt{2}}\cdot e^{-\frac{1}{2}\cdot {\left(\frac{x-\mu_{test}}{\sigma_{test}}\right)}^{2}} \end{array}$$

As can be seen, the spread in the probability density is greatly reduced by a smaller standard deviation. In this case, the resultant confidence of a pass is approximately 86%. Now that both results have been determined, a product distribution can be generated for the combination of the results [7]:(22)$$\begin{array}{c} \phi_{product}\left(x\right)=\frac{\phi_{sim}\left(x\right)\cdot \phi_{test}\left(x\right)}{\int_{-\infty }^{\infty }\phi_{sim}\left(x\right)\cdot \phi_{test}\left(x\right)\cdot dx} \end{array}$$
(23)$$\begin{array}{c} P\left(req.passed\right)=\int_{req}^{\infty }\left(\phi_{product}\left(x\right)\right)\cdot dx \end{array}$$

Applying this method to the results of this example, the product distribution can be determined as below and the resultant confidence in the requirement having passed is 91.82%. Alternatively, it could be said that a peak power of 100.1 kW was achieved with 90% confidence (Figure 8).

It should be noted that neither of the example results considered here were sufficient in and of themselves to provide the requisite validation for this requirement. Only the combination of the results could provide sufficient confidence in the result. Additionally, if the test result confidence was not evaluated online, at the testbed, there is a risk that the sample might be removed or disassembled. Such an action would significantly increase the time and effort needed to finalise the target validation confidence.

The approach suggested here can be demonstrated with a randomised sampling approach in which a priori knowledge is applied with a random component, such as a Monte-Carlo simulation of random input variables fitting a distribution. Assuming a distribution as the product and randomly selecting results from within it to act as the midpoints of the contributing distributions, a series of randomly generated product distributions can be made. Analysing these randomly generated product distributions, the average result from a large number of random simulations can be seen to converge to the a priori distribution, confirming that the product method is suitable for such an analysis.

There are circumstances when it may appear detrimental to the validation process to perform this confidence assessment of the validation target. Changing the parameters of the example above to a condition where the simulation provides lower confidence of a pass, the resultant product distribution is shifted towards the requirement boundary and the confidence in a pass is lower that the testing would provide alone (Figure 9).

In this result, the simulation provides 54.42% confidence, the physical testing provides 86.43%, and the product provides 84.51%. Essentially, this illustrates that, despite two “pass” results, the product was less than the maximum of the individual contributions. This is not a case of the product distribution confidence or of the individual contribution being “wrong”, but of the additional evidence resulting in a decrease in confidence due to additional information being included in the validation assessment.

Similarly, it may not be reasonable to expect a validation in light of anomalous results. If a simulation is conducted and results indicate a failure to meet the target requirement by some significant margin, there may not be an effective testing method to overcome this. In this case, validation can only be achieved by identifying the result as an outlier and disregarding it from the validation (Figure 10).

Finally, thought should be given to the weighting of certain results in the process. The presented example considers an even weighting of the results from the contributing factors. Ideally, the weighting should always be accounted for in the distribution of uncertainties for each method. Alternatively, when more than two sources contribute to the product distribution. This product combination allows for a minority report approach to evaluate the results, in which the correlation of similar results provides a significant boost to the position of the product distribution.

The application of automated testing and an understanding of the core validation targets can allow a highly effective combination at the testbed. If it is assumed that simulations have been conducted in advance of the test but gave insufficient confidence to allow a complete validation in isolation, the testing repeats needed to achieve a confidence target for validation can be determined live at the testbed in response to the results recorded. Such automation could significantly reduce the testing time while also ensuring that validation is consistently achieved.

A similar approach was demonstrated mathematically through the application of Bayesian priors [7]. The Bayesian prior function is given as
(24)$$\begin{array}{c} P\left(A|B\right)=\frac{P(B|A)\times P(A)}{P(B)} \end{array}$$
where P(A|B) is the probability that *A* will occur, given that *B* has occurred and P(A) is the probability that *A* will occur. In this formulation, P(A|B) is the Bayesian posterior, which, for this application, is the confidence in the final validation. P(A) is the Bayesian prior; here, this is the simulation result. P(B|A) is the conditional probability and P(B) is the marginal probability.

As the confidence in the requirement coverage is the objective, the probability functions in the Bayesian priors must be given as Probability Density Functions (PDFs) of the confidence. Krolo and Bertsche [7] apply Beta distributions in application to durability targets and express the marginal probability as an integral function (as the Beta distribution is fitted from 0 to 1):(25)$$\begin{array}{c} f\left(A|B\right)=\frac{f(B|A)\times f(A)}{\int_{0}^{1}\left(f\left(B|A\right)\times f\left(A\right)\right)\times dA} \end{array}$$

An artificial degree of confidence can be applied to a prior in addition to the spread or uncertainty in the result. For example, an early development phase prototype sample result might be considered as a prior to a later sample phase of validation in such a scenario; the confidence in the prior result could be considered in application to the later phase. Taking a Beta distribution, a factor δ can be included to adjust the contribution of the prior in the posterior validation confidence:(26)$$\begin{array}{c} f\left(A\right)=\frac{1}{\beta (\delta A,\delta (B-1))}P{\left(A\right)}^{\delta A-1}P{\left(1-A\right)}^{\delta (B-1)} \end{array}$$

In the case of the application of a subjective prior, such as the carry-over from an earlier development phase, the factor δ is applied by engineering judgement; otherwise, the degree of confidence should be suitably quantified [8]. In the case of the application of a simulation result, it is possible to consider the degree of confidence in the result from past correlations of results. Considering multiple development phases for a product, it would be effective and beneficial to use the earlier development phases to calibrate any simulations and provide a suitable basis for determining the simulation result confidence from the correlation.

## 3. Results

Determining the uncertainty associated with the individual measurements in the validation of an efficiency map at each measurement position, a map of the uncertainty can be applied as an overlay on the mapped area. In doing so, it can be seen that the range of uncertainties shows a strong torque dependence with a particularly high uncertainty in the case of efficiency in the low-torque region. This is primarily due to the calibration of the torque transducer to the full-torque range, resulting in potential offsets becoming a significant proportion at a low load (Figure 11).

## 4. Discussion

Based on the resultant efficiency map uncertainties from a single measurement, the region of the map typically associated with peak efficiency is associated with approximately 2% uncertainty if a 95% confidence in the result is required. The peak efficiency performance is often included as an EDU system requirement and the peak efficiency is typically a design constraining factor that is of high significance due to the implications for powertrain size, weight, cost, and range. Certainly, a 2% uncertainty in validation has the potential to result in significant issues for the design release and the potential for incorrect interpretation.

In the mitigation of such a significant uncertainty, several possibilities present themselves. As discussed, simulation-supported prior information on the EDU performance could be used to collaborate the results, boosting the confidence and reducing the effective uncertainty. Additionally, repeat measurements could be applied to reduce the uncertainty magnitude. The effect of repeating measurements is partially dependent on the individual results achieved. As with simulation-supported validation in the application of Bayesian priors, the simulations could support the physical test findings or contradict them. It is a matter of result integrity that both scenarios are equally considered unless reasons can be found for the discrepancy.

Assuming that multiple physical measurements are taken independently, the convolution of the probability distributions with the mean values of the independent measurements and the standard deviation of the standard errors can be combined as long as the tests are conducted as independent measurements. Taking such an approach, the standard deviation of the combined result can be significantly reduced. Ideally, such an approach is reactively applied within a validation campaign, adjusting the required number of repeats on the basis of the results and the requirement coverage. This should not be applied blindly, as the result of the testing could simply be that the requirement is not met by the design and sampling further only wastes efforts. Instead, a convergence in the confidence of the validation of the requirements might be considered, with the confidence of a pass or fail cut-off being applied at a suitable level. In early development phases, a confidence of 80% may be sufficient to suspend testing while a confidence of 95% may be needed in the final validation phase.

Of additional consideration is the value of individual measurement points. The efficiency map may be of interest overall; however, it may be that specific points on the map are of greater significance to the development and validation process. As such, repeat testing might focus on such points, reducing the overall testing efforts. An assumption of continuous offset might be made to adjust the overall map of the highly correlated points, although this has some potential to introduce unseen errors resulting from the assumption.

Where simulations are to be applied to the validation process, the potential to calibrate a model and quantify the uncertainty in the simulation process is presented through the consideration of multiple development phases. If initial prototype sample phases are additionally used to calibrate simulation models, the calibrated model uncertainty can be determined from the scatter between simulation and measured results. Assuming that design updates are correctly captured by the simulation model or applying a degree of additional uncertainty, such as the factor δ, the reuse of simulations in later development phases can be made effectively [56].

Finally, the calibration of sensors to the specific range of a measurement can offer an additional improvement in accuracy for the result. While this may improve the accuracy of a measurement, it is unlikely to be greatly beneficial where testing such as efficiency mapping is concerned, as either changes in measurement equipment or recalibration would be needed to complete the range, increasing validation efforts overall. Perhaps the most reasonable suggestion for efficiency mapping would be the use of torque sensors calibrated to a lower range for the low-torque region of the efficiency map.

## 5. Conclusions

A case study was presented for the evaluation of efficiency measurement uncertainty from a state-of-the-art test facility for EDU validation. The system of sensors and data acquisition results in relatively high levels of uncertainty, where single measurements are made, and these are likely to be considered insufficient given the high degree of importance of EDU efficiency. Validation can be achieved with sufficient accuracy and confidence with repeated measurements but at the cost of additional efforts. In order to minimise the risk of wasted validation efforts, the requirements for the system should be specified with an acceptable confidence bound, and reactive testing should be applied to quickly evaluate results and adapt the validation plan to remove unnecessary testing. This could be additionally supported by centrally collecting all forms of results and prior information to ensure an accurate and up-to-date validation status at all times.

The addition of simulation results and the convolution of the uncertainty distributions can provide an increase in the confidence of the validation result where the simulations and tests are in agreement. This was demonstrated mathematically through the application of a Bayesian prior method. Such simulation-based validation approaches are of most value when coupled with a calibration campaign for the specific product in an early development phase, allowing the quantification of the result uncertainty. Following this, priors can be combined from multiple sources: simulation, past development phases, similar products, field data, etc.

Finally, repeat testing can be applied to significantly reduce the uncertainty by allowing convolution of the PDFs of the various results; this can greatly converge the distribution outcome and provide a low final uncertainty, which will centre on the mean value of the results.

While this study has focused on efficiency results, EDU requirements can contain a multitude of requirements that fall on a continuous scale and for which uncertainty should be considered in the validation process, including NVH, peak and continuous power, torque response times, etc. As a result, there is potentially a wide-ranging set of sensor and data acquisition systems, which could be evaluated and considered in the overall validation of the system.

## Figures and Tables

**Figure 1 sensors-24-04867-f001:**
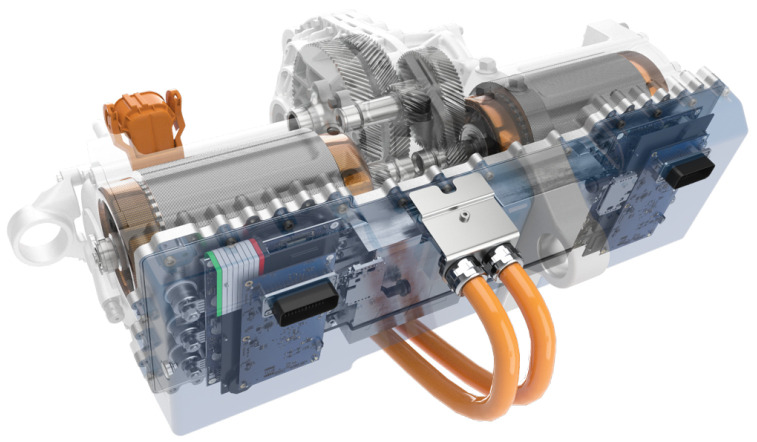
Twin e-Motor electric drive system [2].

**Figure 2 sensors-24-04867-f002:**
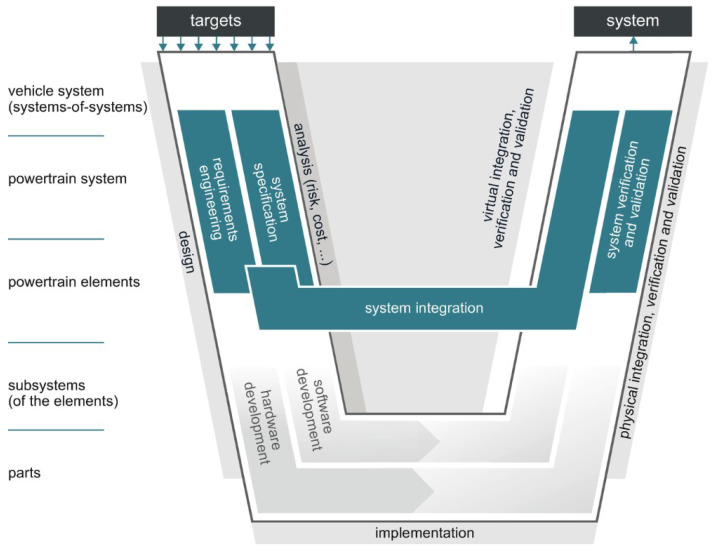
V-model example for systems engineering approach [3].

**Figure 3 sensors-24-04867-f003:**
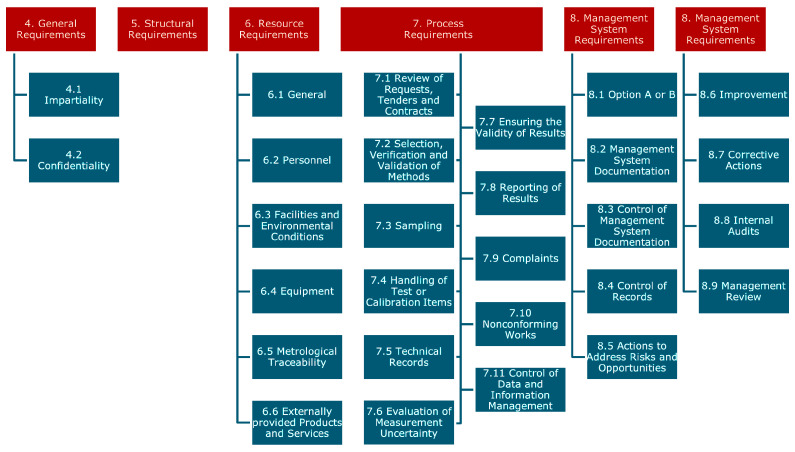
ISO 17025 high-level structure [19].

**Figure 4 sensors-24-04867-f004:**
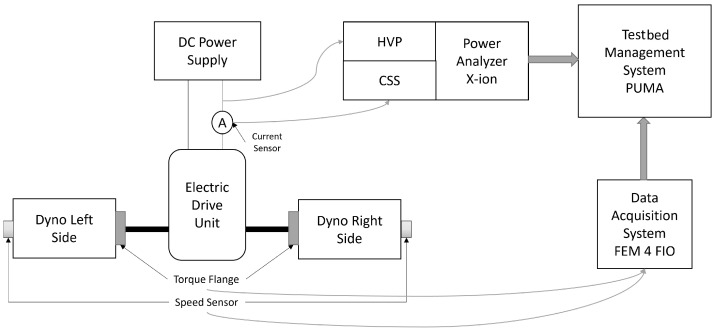
Measurement Chain for EDU Efficiency Testing at AVL.

**Figure 5 sensors-24-04867-f005:**
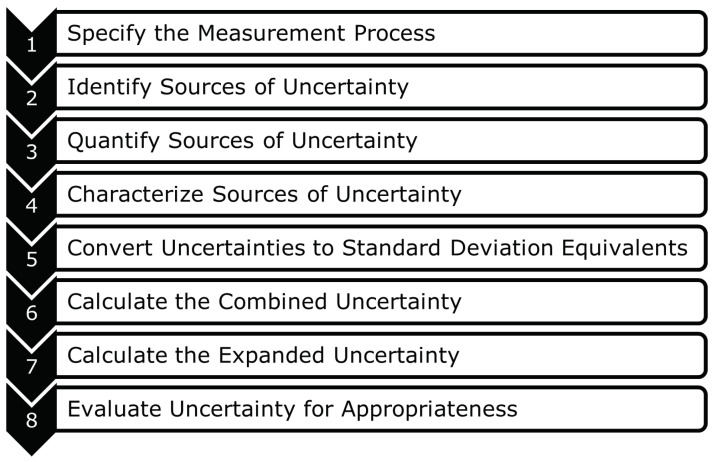
Steps to calculate measurement uncertainty [30].

**Figure 6 sensors-24-04867-f006:**
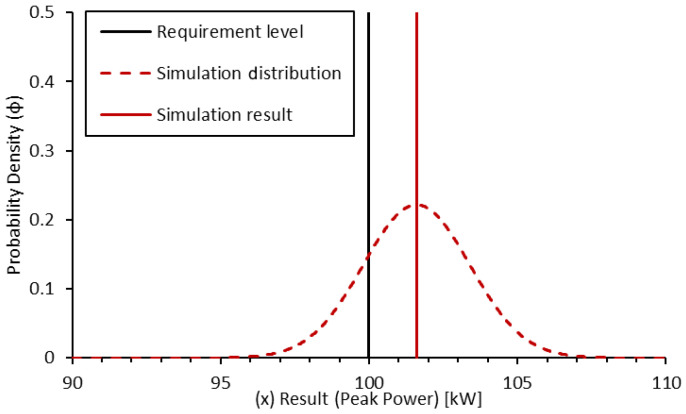
Probability density function for simulation-based confidence in peak power validation.

**Figure 7 sensors-24-04867-f007:**
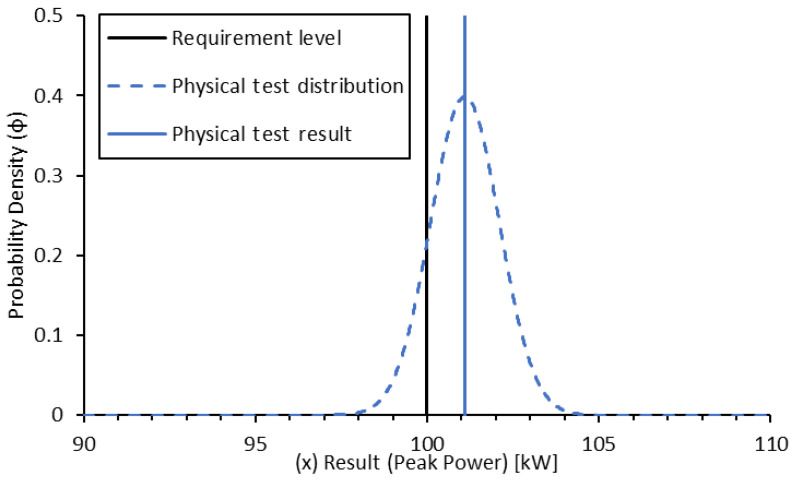
Probability density function for physical test-based confidence in peak power validation.

**Figure 8 sensors-24-04867-f008:**
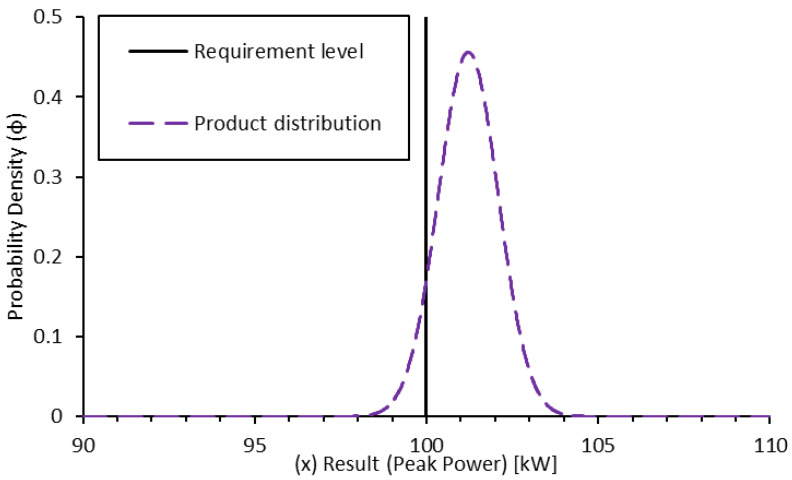
Probability density function for the product of simulation and physical test-based confidence in peak power validation.

**Figure 9 sensors-24-04867-f009:**
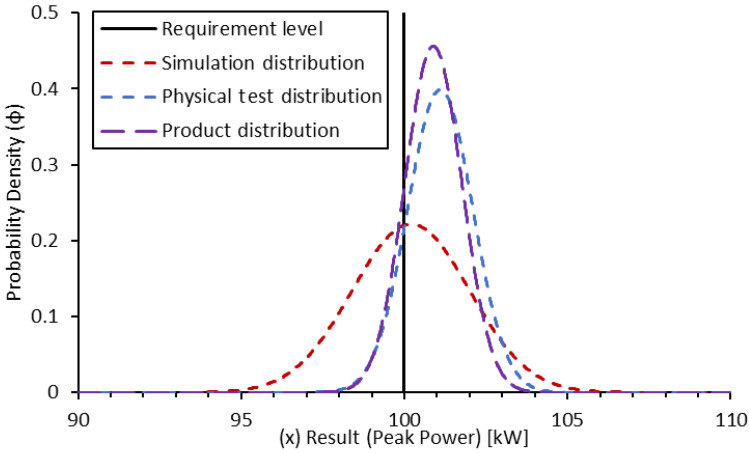
Probability density function for the product of simulation and physical test-based confidence in peak power validation with individual contributions overlaid (simulation result is validation-neutral).

**Figure 10 sensors-24-04867-f010:**
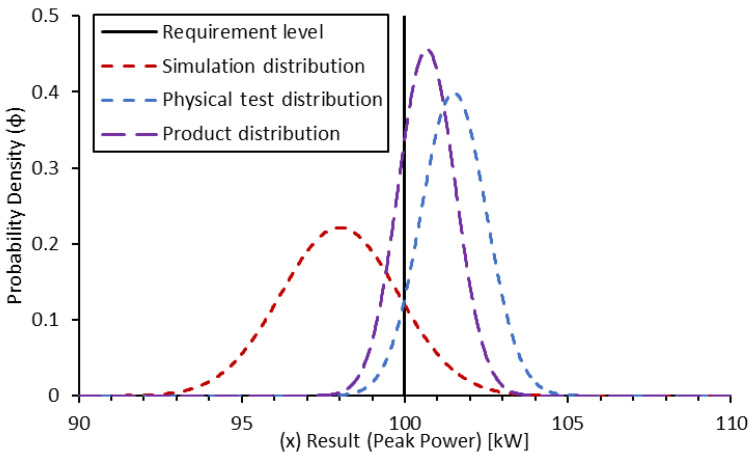
Probability density function for the product of simulation and physical test-based confidence in peak power validation with individual contributions overlaid (simulation result is validation detrimental).

**Figure 11 sensors-24-04867-f011:**
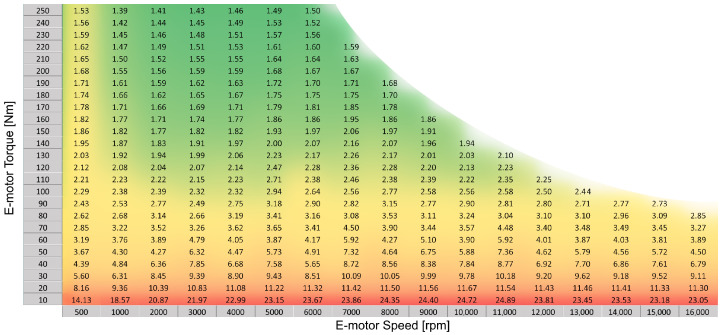
Map of efficiency percentage uncertainty across measurement points considering a 95% confidence interval.

**Table 1 sensors-24-04867-t001:** Coverage factors.

Confidence (%)	Theoretical Coverage Factor
68.27	1.000
70	1.036
80	1.282
90	1.645
95	1.960
95.45	2.000
98	2.326
99	2.576
99.73	3.000

**Table 2 sensors-24-04867-t002:** Torque sensor contributing factors and uncertainty values for HBM T12HP [55].

Contributing Factor	Output Type	Uncertainty Contribution
Sensitivity tolerance	Frequency	±0.05%
Linearity and hysteresis	Frequency	
	0%–20% Mnom	<±0.003%
	20%–60% Mnom	<±0.005%
	60%–100% Mnom	<±0.007%
Temperature effect on zero signal	Field-buses and Frequency	±0.02%
Temperature effect on span	Field-buses and Frequency	±0.005%
Repeatability	Frequency	±0.005%
Parasitic extraneous (off-axis) loading	Axial Limit Force	±39 kN
	Lateral Limit Force	±9 kN
	Bending Limit Moment	±560 Nm

**Table 3 sensors-24-04867-t003:** Rotational speed sensor contributing factors and uncertainty values [52].

Contributing Factor	Output Type	Uncertainty Contribution
Temperature effect on zero signal	Frequency	±0.03%
Temperature effect on span	Frequency	±0.03%
Linearity	Frequency	±0.03%
Hysteresis	Frequency	±0.55%
Turbulence	Frequency	±0.03%

## Data Availability

The data presented in this study are available on request.

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
