# Peer review of "An Application of Uncertainty Quantification to Efficiency Measurements and Validating Requirements through Correlating Simulation and Physical Testing Results†"

_sensors, 2024, doi:10.3390/s24154867_

Round 1

Reviewer 1 Report

Comments and Suggestions for Authors

Dear authors, 

the paper "Application of Uncertainty Quantification to Efficiency Measurements and Validating Requirements through Correlating Simulation and Physical Testing Results" presents an overview of characterizing the overall uncertainty quantification by correlating simulation and experimental results. However, some issues arise when reading the text. Please, respond to the following demands: 

1) The main purpose/contribution of the paper is not well described. I recommend to write a paragraph at the end of the introduction detailing it. 

2) The abstract should be reviewed, making the contribution of the paper more evident. 

3) The first phrase of the conclusion is written that "A case study was presented for the evaluation of efficiency measurent uncertainty...". As the uncertainty quantification was demonstrated in a case study, the term "case study" should be incorporated in the title of the paper. A suggestion might be "Application of Uncertainty Quantification to Efficiency Measurements and Validating Requirements through Correlating Simulation and Physical Testing Results: a case study in a test facility for EDU validation". 

4) Line 69 - "from primary HVDC energy (electrical energy)" - HVDC is a term more frequently used for high voltage transmission lines. It is more convenient to use DC energy instead. 

5) line 71 to 73 - "The TCO for battery electric passenger cars was shown to fall below that of Internal Combustion (IC) vehicles after a period of approximately 7 years in 2021" - Does this number take into account the useful life of EV batteries? 

6) Equation 17 - uncertainty in efficiency - there is a summation of U_inputpower^2 twice. Would the second term be U_outputpower instead of U_inputpower?

Author Response

Dear Reviewer 1,

Thank you for the review and suggested improvements. The authors have tried to include all the suggested points to enhance the quality of the paper. The authors are very grateful for your time in identifying point for improvement of the paper.

1 – An additional paragraph has been added to the end of the introduction indicating some of the key work in the field and highlighting the novelty in this publication.

2 – The authors believe that this is covered by the additional paragraph at the end of the introduction.

3 – The authors feel that the title may become excessively long and have adapted the title to a singular application to attempt to reflect the limit range of applications investigated in the paper; “An Application of Uncertainty Quantification to Efficiency Measurements and Validating Requirements through Correlating Simulation and Physical Testing Results“

4 – High voltage is rated differently in different industries. Within automotive, above 75V DC is within the EU regulations for the level of protection (Directive 2014/35/EU), while in the US, the National Electrical Code suggests a limit at 50V. For this reason, some hybrid designs electrical automotive propulsion have targeted a 48V level to avoid additional regulations. The DC battery inputs in automotive have so far stayed below a 1000V level at which many additional regulation become applicable.
The high voltage direct current term is often applied in automotive to distinguish it from the low voltage (typically 12V) control side of the power electronics and from the alternating current supplied typically supplied to the electric motor.
The acronym has been expanded in the text and a line has been added to clarify the usage in the case.

5 – The study cited here does not consider the battery degradation and notes in the introduction the “This study excludes vehicle resale/ depreciation and battery replacement costs which usually occur in the later stage of ownership.” However, the later referenced commercial vehicle total cost of ownership papers do consider the battery degradation; “Another key finding of this work is that battery efficiency degradation has a minimal impact on the TCO.“ (Wang et al, "A total cost of ownership analysis of zero emission powertrain solutions for the heavy goods vehicle sector")
A statement to this effect has been added to the relevant section.

6 – Thank you for pointing out this typo, it has been corrected in the paper.

Reviewer 2 Report

Comments and Suggestions for Authors

Michael Leighton, Uday Akasapu Application of Uncertainty Quantification to Efficiency Measurements and Validating Requirements through Correlating Simulation and Physical Testing Results This paper is an extended version of our paper published in Assessment of the Efficiency Measurement Uncertainty and the Impact on Validation for Electric Drive Systems presented at 2022 IEEE International Workshop on Metrology for Automotive, Italy, 04-06 July 2022 [4]. Dear Authors, The summary emphasizes the importance of validation in the product development process, as it allows the achievement of design goals with a very low risk of time and financial failure. The Authors propose the use of Uncertainty Quantification (UQ) to test an Electric Drive Unit (EDU). After assessing the measurement chain for testing EDU performance, elementary uncertainties are detected, which in the case of erroneous measurements are the basis for validating the test results. Section 1. Introduction This section presents the construction of a typical EDU used in a Battery Electric Vehicle (BEV) - (Figure 1). Meanwhile, Figure 2 presents a V-model example for systems engineering approach based on the literature [3]. This section is very important because it is mainly devoted to pointing out the dangers that await researchers when using various validation alternatives. With the traditional method of requirements validation, i.e. by quantitatively taking into account evidence collected during physical tests, it is possible to omit the risk of measurement uncertainty. The Authors indicate an alternative to this method of determining measurement uncertainty, which is to repeat measurements with a selection of samples. Of course, all of them must meet the test criteria or requirement threshold. Sample selection increases compliance with the binomial distribution, ensuring that the correct number of samples is selected. The second alternative is the separate validation of system elements shown in Figure 2. Unfortunately, despite the advantages, this approach also introduces risks as evidence is not discovered quickly (a large waste of time). In this way, the Authors demonstrated that only the use of simulation will allow obtaining feedback on the basis of which the development can be modified or interrupted. The above considerations were the basis for selecting the method used in this manuscript. Section 1. Materials and Methods Subsection 2.1 Technical Background Sub-subsection 2.1.1. None, but it should be here. Important data regarding the method and instruments for carrying out measurements are given here. No explanation of the abbreviation HVDC, although it is included in Abbreviations. Others are included in the text and in Abbreviations. Important data regarding the method and instruments for carrying out measurements are given here. No explanation of the abbreviation HVDC, although it is included in Abbreviations. Others are included in the text and in Abbreviations. Sub-subsection 2.1.1. (should be 2.1.2). State of the Art This is a very important sub-section because, based on a detailed analysis of the literature or ISO standards, requirements for tests and laboratories are provided. Sub-subsection 2.1.2. (should be 2.1.3). Efficiency Measurement in EDU The chain n for EDU Efficiency Testing at AVL is discussed here, supplemented with a clear Figure 4. Sub-subsection 2.1.3. (should be 2.1.4). Literature Review An extensive and detailed discussion of the literature on measuring (determining) Uncertainty is provided here, along with Figure 5 (Steps to calculate Measurement Uncertainty according to [5]). Three formulas for uncertainty and Table 1 with Coverage Factors are also provided. Subsection 2.2. Methods Sub-subsection 2.2.1. Associated Uncertainties with Efficiency Measurement Important data regarding the method and instruments for carrying out measurements are provided here. No explanation of the abbreviation HVDC, although it is included in Abbreviations. Others are included in the text and in Abbreviations. Sub-subsection 2.1.1. (should be 2.1.2). State of the Art This is a very important sub-section because, based on a detailed analysis of the literature or ISO standards, requirements for tests and laboratories are provided. Sub-subsection 2.1.2. (should be 2.1.3). Efficiency Measurement in EDU The chain n for EDU Efficiency Testing at AVL is discussed here, supplemented with a clear Figure 4. Sub-subsection 2.1.3. (should be 2.1.4). Literature Review An extensive and detailed discussion of the literature on measuring (determining) Uncertainty is provided here, along with Figure 5 (Steps to calculate Measurement Uncertainty according to [5]). Three formulas for uncertainty and Table 1 with Coverage Factors are also provided. Subsection 2.2. Methods Sub-subsection 2.2.1. Associated Uncertainties with Efficiency Measurement The Authors emphasize that for each measurement the source of uncertainty must first be determined and then the uncertainty quantified. They list sources within the EDU measurement system for efficiency validation (I quote): • Torque measurement (torque transducer) Sensitivity tolerance, non-linearity and hysteresis, temperature effects on zero signal and span, repeatability, parasitic loads. • Speed measurement (rotational angular encoder) Signal sensitivity to temperature, non-linearity and hysteresis • Current measurement (CSS) CSS accuracy, CSS amplifier error limit • Voltage measurement (HVP) HVP accuracy, HVP amplifier error limit • Data Acquisition (AVL Xion) Absolute frequency error on sampling. They then give two very important formulas (5) and (6) for the uncertainty of the torque and rotational speed Measurements. Also very important are the accuracy specifications for the AVL CSS-Box alone, for a combination of the AVL DAQ connection module and CSS-Box, For DC measurements, for DC signals and sinusoidal AC signals in a frequency range up to 30 kHz for the 10V range. The Authors also included in two tables (in Table 2) - Torque sensor contributing factors and uncertainty values for HBM T12HP and (in Table 3) - Rotational speed sensor contributing factors and uncertainty values (in accordance with the literature [49] and [50], respectively). Sub-subsection 2.2.2. Efficiency Measurement Uncertainty Analysis Here the Authors included formulas for uncertainties of speed and torque (12) to (15), for efficiency of the EDU (16) and uncertainty in efficiency (17). Sub-subsection 2.2.3. Combining Simulation Results with Physical Testing Results Here, the Authors compared in Figure 6 requirement level with simulation distribution and simulation results of Probability density function for simulation based confidence in peak power validation, in Figure 7 requirement level with physical test distribution and with physical test result- Probability density function for physical test based confidence in peak power validation, on Figure 8 requirement level z product distribution - Probability density function for the product of simulation and physical test based confidence, on Figure 9 requirement level z simulation distribution, physical test distribution and product distribution - Probability density function for the product of simulation and physical test based confidence in peak power validation with individual contributions overlaid (simulation result is validation neutral) in peak power validation and in Figure 10 requirement level z simulation distribution, physical test distribution and product distribution - Probability density function for the product of simulation and physical test based confidence in peak power validation with individual contributions overlaid (simulation result is validation). Note: Formula (22) requires an explanation or source. Section 3. Results All uncertainty considerations are explained in a clear map presented in Figure 11. Map of efficiency percentage uncertainty across measurement points considering a 95% confidence. What deserves additional praise is that, in addition to the satisfactory Conclusion, the Authors also prepared an extensive discussion of the problem in section 4. Discussion. During the assessment of the literature review, References literature has already been distinguished. Apart from two minor comments, there are no other major comments. Therefore, the Authors deserve congratulations. They did really interesting research and wrote an interesting paper. Best regards, Reviewer

Author Response

Dear Reviewer 2,

The authors would like to express their gratitude for the time and effort spent on the review and the suggestions made for the improvement of the paper. The authors have read your comments and believe there are 3 points for improvements to be considered with responses;

1 – Section numbering under 2.1 – As the reviewer pointed out, sub-section 2.1.1. was not properly referenced in the paper, this has now been corrected.

2 – HVDC term – The reviewer points out that the HVDC abbreviation was not properly introduced in the text, this has been corrected.

3 – Equation 22 – The reviewer points out that the reference for equation 22 was not given in the text and this has now been added.

Round 2

Reviewer 1 Report

Comments and Suggestions for Authors

Dear authors, 

all reviewer's demands have been answered.